# Retinol Levels and Severity of Patients with COVID-19

**DOI:** 10.3390/nu15214642

**Published:** 2023-11-01

**Authors:** Maria Clara da Cruz Carvalho, Júlia Kaline Carvalho Pereira Araujo, Ana Gabriella Costa Lemos da Silva, Nayara Sousa da Silva, Nathalia Kelly de Araújo, Andre Ducati Luchessi, Karla Danielly da Silva Ribeiro, Vivian Nogueira Silbiger

**Affiliations:** 1Graduate Program in Pharmaceutical Science, Federal University of Rio Grande do Norte, Natal 59078-900, RN, Brazil; 2Undergraduate Program of Nutrition, Federal University of Rio Grande do Norte, Natal 59078-900, RN, Brazil; julia.araujo.131@ufrn.edu.br (J.K.C.P.A.); vivian.silbiger@ufrn.br (V.N.S.); 3Graduate Program in Public Health, Federal University of Rio Grande do Norte, Natal 59078-900, RN, Brazil; gabriella_lemos_06@yahoo.com.br; 4Department of Clinical and Toxicological Analyses, Federal University of Rio Grande do Norte, Natal 59078-900, RN, Brazil; nay.sous@gmail.com; 5Graduate Program of Chemistry, Chemistry Institute, Federal University of Rio Grande do Norte, Natal 59078-970, RN, Brazil; 6Department of Nutrition, Center for Health Sciences, Federal University of Rio Grande do Norte, Natal 59078-900, RN, Brazil; 7Graduate Program in Nutrition, Center for Health Sciences, Federal University of Rio Grande do Norte, Natal 59078-900, RN, Brazil

**Keywords:** COVID-19, SARS-CoV-2 infection, retinol, vitamin A, long COVID

## Abstract

The new coronavirus infection represents a serious threat to global health and economies. In this sense, it is paramount to know the nutritional factors that may be related to the prognosis of the disease. Evidence shows that vitamin A may play an important preventive and therapeutic role in supporting respiratory infections as in COVID-19. The aim of our study was to evaluate the association of vitamin A (retinol) status with the prognosis of the disease. A case–control study from a cohort study was conducted in Brazil between May and October 2020. The study population was chosen by convenience, consisting of participants diagnosed with COVID-19. Recruitment was carried out using different approaches, including through dissemination on social media and in four hospitals in the city of Natal/RN, Brazil, recruiting participants from the COVID-19 ward and hospitalized participants who tested positive for the disease. The participants were allocated into two groups according to severity, with a group of mild (*n* = 88) or critical (*n* = 106) patients and compared to a control group (selected before the pandemic, *n* = 46). The extraction of retinol serum was performed and analyzed using the high-performance liquid chromatography method (HPLC). The retinol level was calculated in mmol/L, and levels below 0.7 μmol/L (20 µg/dL) were considered to be a vitamin A deficiency. Our findings suggest that the participants with mild and critical COVID-19 had lower retinol levels compared to the healthy controls (*p* = 0.03). In addition, milder cases of COVID-19 were associated with increased symptoms and prolonged symptoms after 90 days since the beginning of infection. However, the survival analysis showed no association with higher cases of death among participants with vitamin A deficiency (*p* = 0.509). More studies are needed to understand how nutritional status, including vitamin A levels, can influence prognosis and is a risk factor for the development of long COVID syndrome.

## 1. Introduction

A new coronavirus was reported by researchers in Wuhan City, Hubei Province in China. On 30 January 2020, the World Health Organization declared the SARS-CoV-2 epidemic a health emergency of international concern [1]. It subsequently spread around the world quickly because it is a highly contagious disease, and since then, more than 6 million people have died from the disease, becoming a threat to public health. The clinical status of patients varies from an asymptomatic, mild upper respiratory tract infection (URTI), to severe viral pneumonia, acute respiratory distress syndrome (ARDS), and death [2].

Although vaccination has been effective in significantly reducing the development of severe cases of the disease, it does not prevent infection. In addition, the need for booster doses is still being investigated. Part of the population has been shown to be resistant to immunizations, and the specific antiviral medication is controversial. In addition, the emergence of new variants threatens global health and economies [3].

Moreover, management of the patient after discharge is still a challenge, since many patients persist with symptoms 4 weeks after the onset of the disease, being characterized in the literature as “post-COVID syndrome” or “long COVID”. The most common symptoms include anosmia, fatigue, chest pain, hair loss, difficulty concentrating, and sexual dysfunction [4,5]. The etiology of “long COVID” is still unclear, as some authors attribute the effect of SARS-CoV-2 to the development of the syndrome, and other authors to the biopsychosocial effects of the disease [6].

The role of nutritional status and diet are widely discussed in the literature given their importance in controlling and treating infections. In the context of COVID-19, nutrients have been associated with decreased cytokine storm due to their anti-inflammatory capacity [7]. Evidence shows that vitamin A can play an important preventive and therapeutic role in supporting respiratory infections. 

The biological role of retinoids is widely discussed, including the maintenance of vision and health, epithelial and mucosal regulation, bone metabolism, and antioxidant properties [8]. Vitamin A plays a crucial role in certain immunoregulatory processes, since it promotes the proliferation of T lymphocytes (by increasing IL-2) and also their differentiation, especially in regulatory T cells [9]. Vitamin A can significantly mediate oxidative damage due to its antioxidant capacity, thus contributing to lung regeneration in patients with COVID-19 [10]. Furthermore, vitamin A has been associated with better antibody production in studies involving vaccination against infectious diseases, such as malaria and polio [11,12].

Vitamin A has been shown to be a potential therapeutic target for COVID-19 through mechanisms including the inhibition of inflammatory responses and biological processes associated with reactive oxygen species, possibly binding to fatty acids in the spike protein in SARS-CoV-2, inhibiting its entry into the cell and signaling gene expression, which induces the neutralization of signals associated with COVID-19 [13,14,15,16,17,18]. In addition, the literature suggests a special role of a gene inducible by retinoic acid (RIG-I) in managing viral infection. These mechanisms include the role of RIG-1 as a suppressor at the beginning of SARS-CoV-2 infection in human lung cells, which can be influenced by the available retinol levels [19,20,21].

Thus, this study characterized a population of mild and critical patients with COVID-19 and their retinol levels compared to a control population. Our objective was to evaluate the association of vitamin A (retinol) status with disease prognosis and discuss possible explanations for the associations found.

## 2. Materials and Methods

### 2.1. Study Population

This case–control study was conducted between May/2020 and October/2020. The study population was chosen by convenience and composed of adults and older adults (from 18 years of age) of both sexes, diagnosed with COVID-19 through polymerase chain reaction (PCR) or serology performed by immunofluorescence. Recruitment was carried out using different approaches: the first was through dissemination on social media (such as Instagram, WhatsApp, Telegram), local media (such as television programs, television newspapers), and in 4 hospitals in the city of Natal/RN, Brazil, recruiting participants from the COVID-19 ward and hospitalized participants who tested positive for the disease. Participants were grouped by disease severity according to the National Institutes of Health (2022) criteria: mild when presenting any of the signs and symptoms of COVID-19, but who did not have shortness of breath, dyspnea, or abnormal chest imaging; critical cases were considered as those with oxygen saturation levels (SpO_2_) < 94%, the arterial partial oxygen pressure to the fraction of inspired oxygen (PaO_2_/FiO_2_) ratio < 300 mm Hg, a respiratory rate >30 breaths/min, or lung infiltrates >50%, and/or respiratory failure, septic shock, and/or multiple organ dysfunction [22]. Participants who had their biological samples compromised (due to the impossibility of extraction and/or insufficient amount of serum), who used nutritional supplements, who dropped out of the study, or who had invalid telephone contact were excluded from the study. Thus, a total of 194 participants, with 88 mild and 106 critical patients were included in the study, as shown in Figure 1.

### 2.2. Data Collection and Characterization of the Population

Participants or their responsible family members answered a questionnaire, personally or by phone, about their personal data, socioeconomic, ethnicity, symptoms, comorbidities, hospitalizations, drugs used before and during infection, and supplements and complications associated with the disease. 

Blood collection of the hospitalized participants occurred in a hospital environment at the time they were hospitalized. This occurred from 15 to 20 days after diagnosis for mild patients, immediately after isolation. Blood samples were taken in the morning by trained professionals with the participants fasting for at least 8 h.

The participants were contacted again 90 days after the first application of the questionnaire in order to assess the persistence of symptoms and survival. Nutritional status was determined by obtaining the body mass index (BMI) from self-reported values by the study participants or from the electronic medical records of the hospitals. 

The control group came from the database of a previous study by our research group [23], consisting of participants recruited from a hemodynamics unit, selected for undergoing their first coronary angiography; however, only those who obtained a negative result for the diagnosis of coronary arterial disease were selected. In addition, none of these participants had any condition that affected vitamin A levels or could cause conflicts regarding the analysis of retinol for comparative purposes. Data collections were carried out prior to 2019 (before the pandemic), and participants never received a diagnosis of COVID-19. As many participants were asymptomatic for the disease, it would be difficult to know whether they were in fact people who did not have the disease or were just asymptomatic, or if they had not had the disease before. This would make it difficult to create a reliable control group. The participants’ retinol levels were assessed using the same methodology and equipment.

### 2.3. Measurement of Vitamin A

Peripheral blood samples were collected by trained professionals in a home environment or made available by selected hospitals. The samples were allocated in tubes for serology with separator gel and then taken for centrifugation (at room temperature) for 5 min (500× *g*) after thawing to separate the whole blood and the serum. Finally, aliquots containing 500 μL of serum were stored at −80 °C until the biochemical analysis. The retinol serum extraction was performed according to a method adapted from Ortega et al. (1998) [24], and analyzed using the high-performance liquid chromatography method (HPLC) (Shimadzu, Kyoto, Japan) at a wavelength of 325 nm. The retinol level was calculated in mmol/L, and levels below 0.7 μmol/L (20 µg/dL) were considered low, based on the WHO recommendation [25].

### 2.4. Statistical Analysis

The collected data were transferred to the Research Electronic Data Capture (Redcap, Vanderbilt University) input tool and analyzed by the Statistical Package for the Social Sciences (IBM SPSS Statistics version 26). The Kolmogorov–Smirnov test was used to determine the normality of the variables. The continuous variables were represented by the median (interquartile range—IQR) and the categorical variables by the percentage and *n* absolute. We used the Kruskal–Wallis test followed by a post-hoc test to evaluate the differences among the continuous variables. The chi-squared independence test was used to detect associations among the groups of categorical variables. Survival was assessed using Kaplan–Meier survival curves. Differences among the groups were considered statistically significant when the *p*-value was less than 0.05.

## 3. Results

As shown in Table 1, the data from 194 participants with COVID-19 were evaluated and divided into two groups according to the severity scale: participants with mild upper respiratory tract infection (URTI)—called “mild”—with 88 participants; and severe viral pneumonia to acute respiratory distress syndrome (ARDS)—called “critical”—with 106 participants. The data from the control group, with 46 participants, were also evaluated.

No significant differences were found in relation to body mass index (*p* = 0.241) or sex (*p* = 0.973) among the groups, showing homogeneity among the participants regarding these variables. 

A higher median age of 67 (55; 79) years was observed in critical participants, which was significantly higher than in the other groups studied (*p* < 0.001). Similarly, higher ALT, AST, and creatinine values were observed in critical participants, which were significantly different than in the other groups.

As shown in Table 2, significant differences were found (*p* < 0.001) in the numbers of comorbidities in the participants between the groups. Participants with 1 to 2 comorbidities were more frequent among mild cases, totaling 55.7% (49). Among the critical cases, the highest frequency of participants in this group presented 2 to 3 comorbidities, totaling 41.5% (44).

However, the mild participants had more symptoms associated with COVID-19 compared to the critical participants. While the critical participants had mostly 1 to 4 symptoms, representing 67.9% (72) of this group, there was a greater variation in the reported symptoms among the mild cases, since 26.4% (23) had 1 to 4 symptoms, 34.5% (30) had from 5 to 8, and 31.0% (27) had more than 9. There was a significant difference (*p* < 0.001) between the groups.

In addition, participants with critical COVID-19 had a longer duration of illness (*p* < 0.001) compared to the mild patients, with a median of 24 (16, 36) days versus 10 (5, 20) days.

Higher retinol levels were observed in the control group, with a median of 1.85 (1.38; 2.44) mmol/L, compared to the participants with COVID-19, in which median values of 1.20 (0.42; 2.71) mmol/L were found among the mild cases and 1.58 (0.88; 2.38) mmol/L among the critical cases, as shown in Figure 2A. Significant differences were observed in the ROH levels (*p* = 0.036) between the mild participants and the controls (*p* = 0.036). In addition, deficiency prevalence was also found in higher frequency among the mild cases compared to the control group (*p* = 0.001), with 38.60% (34) of participants in this condition.

The median CRP values were 2.70 mg/dL (0.40; 6.00) among the mild participants and 14.05 (5.56; 24.85) mg/dL among the critical participants, as shown in Figure 2B. Although they differed significantly (*p* = 0.01), both groups showed high CRP values (>0.3 mg/dL) according to the reference.

No significant differences (*p* = 0.509) were found in the Kaplan–Meier survival analysis between deaths (outcome) and participants’ retinol status (factor) in the time between the date of diagnosis and contact 90 days later (Figure 3). Nevertheless, it was observed that participants with retinol deficiency tended to have a faster negative outcome (death) compared to those with adequate retinol levels (between 75 and 100 days—blue line; >125 days—green line).

## 4. Discussion

Lower retinol levels were found in the participants with COVID-19 when compared to the controls, and lower retinol levels were also observed among the mild participants. Mild cases also presented a higher number of symptoms, and higher comorbidities, higher median age, lack of hospitalization, and longer time with the disease were observed in the critical participants when compared to the other groups studied.

Older age and a greater number of comorbidities in the group of critically ill participants have already been observed in population studies of COVID participants, since older adult participants and the presence of comorbidities are associated with worse prognosis of the disease, such as a greater need for hospitalization, longer illness time, and higher incidence of death. Studies show a progressive increase in mortality with increasing age, associated with the presence of comorbidities such as obesity, cardiovascular diseases, chronic obstructive pulmonary disease (COPD), immunosuppression, and diabetes [26,27,28,29]. This puts these participants into priority groups of immunization and control strategies, as they belong to the highest-risk group.

Although the mild participants had a lower number of comorbidities compared to the critical cases, a higher frequency of reported symptoms was observed among these participants. Differences in the distribution of initial symptoms between severe cases and mild cases were evaluated in a meta-analysis [30], including studies that added data from 3326 participants. The analysis showed that abdominal pain, dyspnea, anorexia, diarrhea, fatigue, expectoration, fever, and cough were more likely to be reported by critical participants. In contrast, chest tightness, pharyngitis, nausea or vomiting, headache, and myalgia were more reported by participants with mild symptoms.

However, with advancing age, the ability to report the presence of symptoms, and the alteration of perceptions such as the senses (i.e., smell and taste) are diminished by genetic and environmental causes that alter the anatomy and physiology of the systems involved, compromising their ability to feel and report these variations [31,32,33]. In this case, the smallest number of symptoms among older (critical) participants may actually be underreported and not necessarily represent the absence of these symptoms. 

The mild participants were more likely to have persistent symptoms after 90 days from the onset of the disease. The emergence of “long COVID”, a term currently adopted to describe a diverse set of persistent symptoms in participants with COVID-19, is suggested after a minimum of 4 weeks since the onset of infection [4,5]. Findings in the literature show a frequency of persistent symptoms in participants after mild COVID-19 infection of between 10% and 35% of cases. Fatigue was the most persistently reported symptom, followed by dyspnea, cough, chest pain, headache, decreased mental and cognitive status, and olfactory dysfunction [34]. However, no studies evaluating the association of vitamin A nutritional status and post-COVID-19 symptom persistence have been found. The etiology of this syndrome is still under discussion, since some authors attribute the effect of SARS-CoV-2 to the development of the syndrome, and other authors to the biopsychosocial effects of the disease [6].

Although the mechanisms surrounding COVID-19 are not fully understood, in previous epidemics of SARS, H1N1, and Ebola, for example, cases of participants who had symptoms even after the disease have also been reported [35]. The presence of comorbidities in these participants may impact the persistence of these symptoms [36,37].

When compared to the control group, the highest age group was also found among the critical participants. In addition, increased creatinine, AST, and ALT levels were observed in the critical participants, differing significantly from the groups compared. The literature explains a significant association between high creatinine, ALT, and AST levels and the severity of COVID-19. The identification of these clinical laboratory predictors is important for risk stratification and to evaluate disease progression [38,39]. 

The participants were also evaluated for their nutritional status of vitamin A (retinol), where the median ROH values were lower among the mild participants, who also presented a higher deficiency frequency for this micronutrient. Experimental studies have shown that retinoic acid-inducible gene-I (RIG-I) rapidly consumes a large amount of the body’s retinoid reserve for viral identification mechanisms during SARS-CoV-2 infection, leading to decreased plasma retinol levels during infection. These findings also suggest that the serum values of these micronutrients may be reduced during COVID-19 [15,16].

However, overexpression of the SARS-CoV-2 N protein suppresses the activation of retinoic acid-inducing gene type I (RIG-I), preventing interaction between the tripartite protein 25 (TRIM25) and RIG-I and interaction with the DExD/H domain, which prevents activation of this inflammatory pathway. These results demonstrate that a high viral load, as is the case for critical participants, can attenuate activating this receptor, and therefore, maintain plasma vitamin A levels, since they are not being used for the activation of these receptors [16,17,18]. 

Studies have shown a range from 0.18 to 1.68 μmol/L of retinol in patients with COVID-19, for which the blood levels were lower in the patients than the controls, similar to our findings [22,40,41,42,43,44]. However, most studies did not separate by severity of the disease; they only compared the data of patients with COVID-19 with their controls. Only the study by VOELKLE et al. (2022) [41], conducted with 57 adults, compared the plasma retinol levels in mild (n = 42) and critical [15] patients. Contrary to our findings, critical patients presented lower retinol values, with a median of 0.7 (0.4–1.1) μmol/L, compared to mild cases, with a median of 1.5 (1.0–2.0) μmol/L. Nevertheless, the non-exclusion of patients supplemented with vitamins, the dietary profiles among populations, and the lack of a control group could have led to this difference between the results.

In this case, the retinol values in the participants with COVID-19 may be reduced as a result of a decrease in their carrier, and not necessarily because they have decreased baseline values. C-reactive protein (CRP) is a protein synthesized by the liver, and its levels increase in response to inflammation. At the same time, the production of a number of other proteins is reduced, as is the case with retinol-binding protein (RBP). Thus, the decrease in hepatic RBP synthesis may reflect a lower secretion of retinol from hepatic stores, even in situations of adequate hepatic reserves [45].

As limitations of this study, we can highlight the difference in the collection times between the evaluated groups, since the mild participants had their samples collected after isolation, and the critical participants at the time of hospitalization. Another point is the lack of previous information on these participants that could demonstrate the real influence of the infection on the parameters evaluated, as well as the size of our cohort, especially the control group. However, the sample size was sufficient to carry out the statistical analyses presented and create comparative groups. In addition, it was possible to identify that the participants with COVID-19 had significantly different changes according to severity, and also compared to the control group. Additional studies are important to investigate how nutritional status, including vitamin A, may be influencing the persistence of symptoms after COVID-19.

## 5. Conclusions

Our findings show that the COVID-19 participants had lower retinol levels compared to the controls. The critical participants had a higher median age and worse prognosis, such as longer disease time, need for hospitalization, death, and a greater number of comorbidities. In addition, milder cases of COVID-19 were associated with increased symptoms and prolonged symptoms 90 days after the beginning of infection. However, the survival analysis showed no association between higher cases of death among the participants with vitamin A deficiency. More studies are needed to understand how nutritional status, including vitamin A levels, can influence long COVID syndrome.

## Figures and Tables

**Figure 1 nutrients-15-04642-f001:**
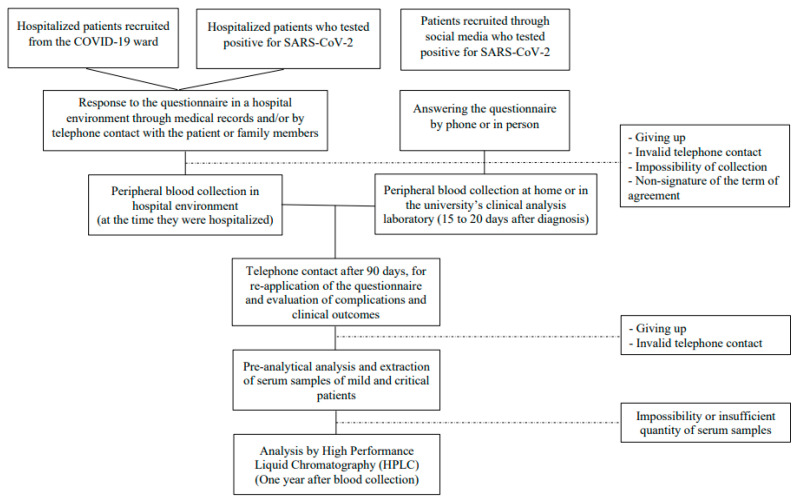
Flowchart of the study design.

**Figure 2 nutrients-15-04642-f002:**
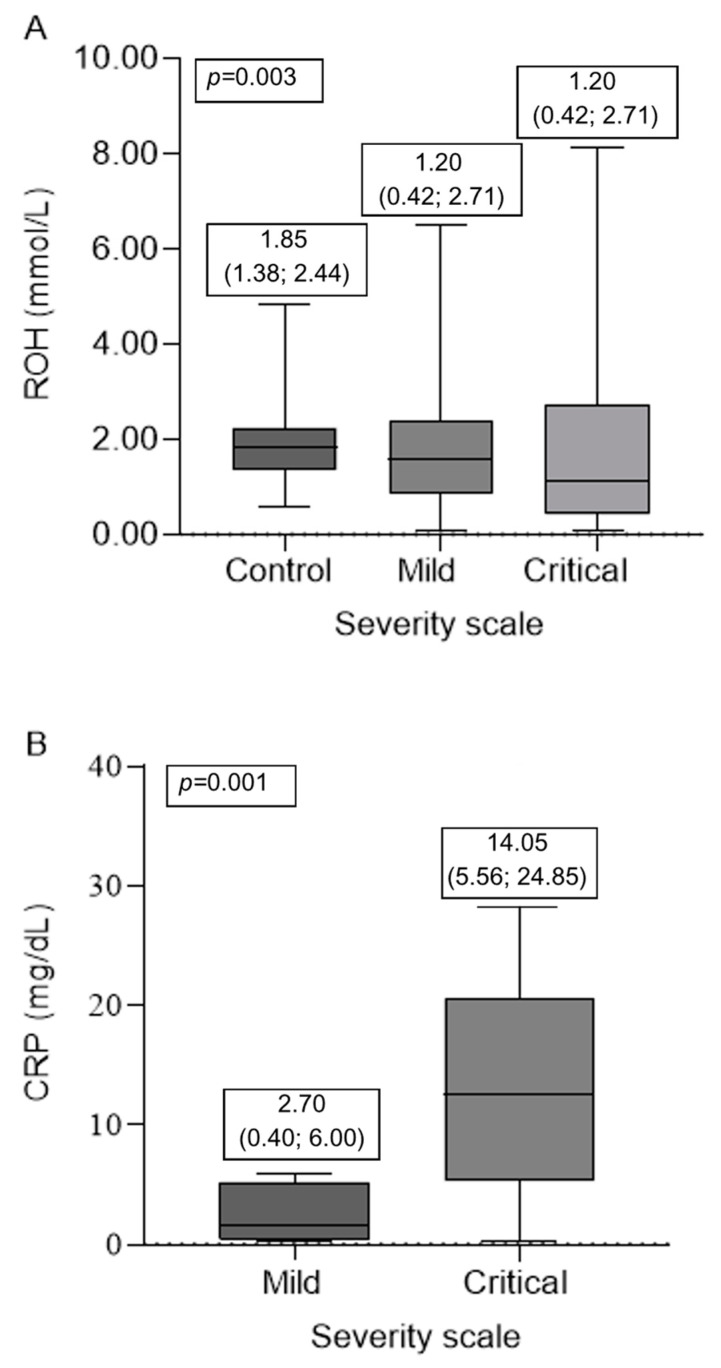
Characteristics of vitamin A (**A**) and CRP nutritional status of critical and mild COVID-19 participants and controls. ROH, retinol; CRP, C-reactive protein. Median between serum retinol (**A**) and C-reactive protein (**B**) levels in the evaluated groups. Retinol values >0.7 mmol/L were considered acceptable, and less than this level as deficient. *p*-values < 0.05 were considered statistically significant.

**Figure 3 nutrients-15-04642-f003:**
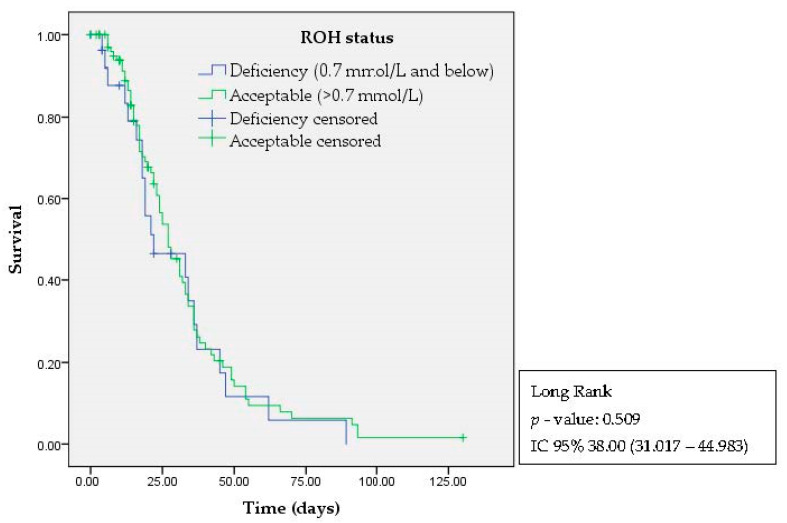
Kaplan–Meier survival curve.

**Table 1 nutrients-15-04642-t001:** Characterization of participants with COVID-19 and controls.

Variables	Controls (*n* = 46)	Severity Scale		*p*-Value
		Mild (*n* = 88)	Critical (*n* = 106)	
	Mean (SD) or Median (IQR) or %(*n*)			
Age (years)	54 (44; 62)	47 (42; 55)	67 (55; 79)	**<0.001**
BMI (kg/m^2^)	29.50 (26.78; 31.48)	27.68 (25.26; 30.48)	28.51 (24.09; 31.69)	0.241
Sex				0.973
Female	56.5% (26)	54.5% (48)	54.7% (58)	
Male	43.5% (20)	45.5% (40)	45.3% (48)	
Creatinine (mg/dL)	0.85 (0.80; 1.00)	0.93 (0.68; 1.10)	1.56 (0.90; 3.07)	**0.001**
AST (U/L)	31.00 (26.00; 41.00)	24.00 (16.00; 52.50)	51.00 (31.00; 68.00)	**<0.001**
ALT (U/L)	25.50 (20.00; 34.00)	33.50 (25.00; 82.50)	44.00 (24.00; 84.00)	**<0.001**
Obesity	34.8% (16)	30.7% (27)	23.6% (25)	0.308
Hypertension	80.4% (37)	37.5% (33)	63.2% (67)	**<0.001**
Diabetes	24.4% (11)	6.8% (6)	32.1% (34)	**<0.001**

BMI, body mass index; kg/m^2^, kilograms (per square meter); mg/dL, milligrams per deciliter; U/L, unit per liter. *p*-values < 0.05 were considered statistically significant and are marked in bold in the table, calculated using the chi-squared independence test and Kruskal–Wallis test.

**Table 2 nutrients-15-04642-t002:** Characterization of Brazilian COVID-19 participants between May/2020 and October/2020.

Associated with COVID-19	Mild (*n* = 88)	Critical (*n* = 106)	*p*-Value
	Median (IQR) or %(*n*)		
Time with the disease (days)	10 (5, 20)	24 (16, 36)	**<0.001**
Comorbidities			**<0.001**
None	33% (29)	16.0% (17)	
1 to 2	55.7% (49)	30.2% (32)	
2 to 3	11.4% (10)	41.5% (44)	
>4	-	12.3% (13)	
Symptoms			**<0.001**
None	8.0% (7)	-	
1 to 4	26.4% (23)	67.9% (72)	
5 to 8	34.5% (30)	14.2% (15)	
>9	31.0% (27)	17.9% (19)	
Survival (90th day)			**<0.001**
No	-	49.5% (52)	
Yes	100% (88)	50.5% (53)	
Persistent symptoms after 90 days			**<0.001**
No	39.8% (35)	92.5% (49)	
Yes	60% (53)	7.5% (4)	

BMI, body mass index; kg/m^2^, kilograms (per square meter); mg/dL, milligrams per deciliter; U/L, unit per liter. *p*-values < 0.05 were considered statistically significant, and are marked in bold, in the table, calculated using the chi-squared independence test.

## Data Availability

The datasets analyzed during the current study are available from the corresponding author upon reasonable request.

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
