# Peer review of "Retinol Levels and Severity of Patients with COVID-19"

_nutrients, 2023, doi:10.3390/nu15214642_

Round 1
Reviewer 1 Report
The manuscript from da C Carvalho et al. reports possible associations between serum retinol levels and COVID-19 severity in healthy subjects (n = 46) and patients experiencing either mild (n = 88) or critical (n = 106) COVID-19 severity. Statistically significant (p = 0.036) lower serum retinol levels were observed only for the mild COVID-19 group compared to healthy patients but not for other group-specific comparisons.
The study appears to be technically sound and the work and data are believable. However, the manuscript has a number of serious weaknesses that detract from the work and manuscript.
The text is very poorly written, especially with regards to English usage, and in places the authors’ intended meanings are not understandable. Some representative examples of this, only in the Discussion section, include:
In the 4th to last paragraph, the last sentence, the authors state “…can attenuate the activation of the receptor, and therefore, keeping the plasma levels counting, since they are not being used…” What do the authors mean by “keeping the plasma levels counting”?
In the next paragraph the first sentence states “…being values close to our findings and inferior to the values of controls.” I would guess that the authors are trying to say that the “…blood levels were lower in patients than controls…” But this is simply a guess. This needs to be made clear.
In the next paragraph and its first sentence, the authors mention that hepatic synthesis of C-reactive protein is diminished in response to inflammation. The second sentence states that synthesis of a number of other hepatic proteins are also reduced including that of retinol-binding protein. I would guess that the authors may be trying to say that the lower retinol levels observed in mild COVID-19 patients may arise from an inflammation reduced reduction in hepatic synthesis of RBP and less secretion of retinol from hepatic stores. But this needs to be clarified in the text; this is how I interpret this text. If this is indeed the intended meaning, I also would note that RBP levels in serum were not measured for these patients and consequently this is speculation, albeit possibly correct speculation. Why were serum RBP levels not measured? These measures would benefit the study.
The entire text needs to be carefully edited by a native speaker or by a manuscript service to make the text and the authors’ intended meanings fully understandable to readers.
A second major concern centers on how the authors interpret their data. The first sentence of the Conclusions section states “Our findings suggest that patients with COVID-19 have decreased levels of retinol compared to healthy controls, and may vary according to severity.” Although a statistically significant difference in serum retinol levels were observed for healthy versus mild COVID-19 patients (p = 0.036), no other statistically significant difference was observed. This does not support well the authors’ contention on this point. Moreover, it is unclear to this reviewer, if this statistically significance difference is truly a physiologically significant one. The serum levels are not quantitatively very different and it is unclear whether this would meaningfully affect disease outcomes.
The authors further state from the Kaplan-Meier survival curve provided as Figure 3 that “…it was observed that patients with retinol deficiency tended to have faster negative outcome (death) compared to those with adequate retinol levels.” For this reviewer, the two curves are essentially identical. I might be missing something that the authors may wish to address in the text.
For this reviewer, it is surprising that larger effects on serum retinol were not detected for the Critical COVID-19 group. The authors need to more carefully consider this point in their discussion and in the other parts of the text which reports or considers their data. Rather than trying to agree with other investigators and/or with what may have been expected in advance, it would be better for the authors to address fully the conclusions that can be reached from their data even if they are negative and unexpected. The text needs to be changed to truly consider and reflect the data and its significance.
Needs to be improved considerably.
Author Response
Dear Editor,
We would like to resubmit a revised version of the manuscript ID nutrients-2551050 (Retinol levels and severity of patients with COVID-19). Please refer to the point-by-point replies to the reviewers’ comments below. We are thankful for the comments and suggestions of the reviewers which we believe have helped refine and improve our manuscript. We hope that the reviewers’ concerns are addressed in the present version of the manuscript. We appreciate the opportunity to resubmit our study to the Nutrients journal.
Thank you for your consideration. I look forward to hearing from you.
Sincerely,
Maria Clara da Cruz Carvalho.

Reviewer 2 Report
This is a short but important description of the correlation of the circulating levels of retinol and the severity of patients with COVID-19. Overall the study is well planned out, the manuscript logically presented and is generally well written (although see later comments on use of English).
More details are needed for some of the methods, for instance how Covid severity is measured e.g. how were the elements of the National Institutes of Health criteria used, what was the expertise of the person who did this, was it a single person etc. Also needed is what time of day and time after last meal was blood collected? Blood is collected 15-20 days after diagnosis for mild and after hospitalization for severe – comment on this difference is needed in the discussion
It is stated that a control group would be difficult to collect during the pandemic because many patients were asymptomatic for the disease, it would be difficult to know whether they were in fact people who did not have the disease or were just asymptomatic. This is not the case as the qPCR test could be applied. The control group consisted of those who had their first coronary angiography which is not ideal as, even though they obtained a negative result, they were presumably requiring the test because of symptoms e.g. 80% had hypertension. The case for the control group should be better made.
In Figure 3 the information on statistics hides part of the graph which is important to see, particularly because of the statement that those “with retinol deficiency tended to have a faster negative outcome (death) compared to those with adequate retinol levels.”. This needs to be better justified.
In the discussion it is stated “higher odds of presenting persistent symptoms after 90 days, especially when they were deficient for retinol levels.” What is the data that supports this? Later it is stated for the difference between the present and the Voelkle study “However, a lower sample size was explained especially in the group of critical patients, which can lead to this difference be-tween the results.” - this can hardly be described as an explanation. More discussion of the differences between the studies are needed. In addition the Sarohan et al Clin Nutr Open Sci. 2022 paper should be listed in addition to their Cell Signal paper.
English needs to be improved in some spots e,g, in introduction p2 states “Vitamin A can significantly mediate oxidative damage” which implies Vitamin A causes oxidative damage, while the next part of the sentence “how much regenerative capacity of the lungs in patients with COVID-19” seems missing words. Or on p6 what is meant by “Retinol value >0.7 mmol/L were considered acceptable, and inferior to it, in deficiency.” The manuscript needs to be read through again for grammatical sense.
Author Response

(The authors gave the same response as above.)
